# National Monitoring for Menstrual Health and Hygiene: Is the Type of Menstrual Material Used Indicative of Needs Across 10 Countries?

**DOI:** 10.3390/ijerph17082633

**Published:** 2020-04-12

**Authors:** Annie D. Smith, Alfred Muli, Kellogg J. Schwab, Julie Hennegan

**Affiliations:** 1Department of Population, Family and Reproductive Health, Johns Hopkins Bloomberg School of Public Health, Baltimore, MD 21205, USA; asmit281@jhu.edu; 2Ruby Cup by Ruby Life, Nairobi 00100, Kenya; alfred@rubycup.com; 3The Water Institute, Department of Environmental Health and Engineering, Johns Hopkins Bloomberg School of Public Health, Baltimore, MD 21205, USA; kschwab1@jhu.edu

**Keywords:** menstrual hygiene, menstrual health, outcome assessment, health indicators, women’s health, reproductive health

## Abstract

Surveys monitoring population health and sanitation are increasingly seeking to monitor menstrual health. In the absence of established indicators, these surveys have most often collected data on the type of menstrual material used. This study investigated whether such data provides a useful indication of women’s menstrual material needs being met. Using data from 12 national or state representative surveys from the Performance Monitoring and Accountability 2020 program, we compared self-reported menstrual material use against respondents’ reported menstrual material needs (including needing clean materials, money, or access to a vendor). The use of menstrual pads did not indicate that menstrual material needs were met for many respondents. Of those exclusively using pads, a pooled 26.4% (95% Confidence Interval 17.1–38.5) of respondents reported that they had unmet material needs. More disadvantaged groups were particularly misrepresented; of rural women exclusively using pads, a pooled 38.5% (95% CI 27.3–51.1) reported unmet material needs, compared to 17.1% (95% CI 12.4–23.0) of urban women. Similar disparities were observed for levels of education and wealth, with a pooled 45.9% (95% CI 29.2–63.6) of women in the lowest wealth quintile reporting unmet material needs. Findings suggest that caution is needed when using menstrual material use as an indicator for menstrual health.

## 1. Introduction

Menstrual health has been increasingly recognized as an important component of reproductive health and gender-sensitive water, sanitation, and hygiene (WASH) needs [1,2]. A growing body of evidence has highlighted the importance of menstrual experiences in the lives of women and girls and the challenges associated with menstruation, particularly in low- and middle-income countries (LMICs) [3,4,5]. A broad range of challenges for menstrual health have been reported, including access to clean absorbent materials; availability of safe, clean, and private spaces for cleaning, changing, and disposing of materials; access to adequate menstrual and reproductive health education; insufficient diagnosis and treatment of menstrual disorders; and socio-cultural norms that stigmatize menstruation and limit social support [1,3,4]. In response, an increasing number of policies and programs to improve menstrual health have emerged, often focused on increasing access to commercial menstrual materials, particularly menstrual pads. Kenya was one of the first countries to remove value-added tax (VAT) on disposable menstrual product imports, and has since been joined by a number of other countries eliminating VAT or sales taxes, including Rwanda, India, South Africa, Canada, Germany, and several U.S. cities and states [6,7,8,9]. Further, countries including Kenya, Scotland, India, and Nepal have committed to national distribution of menstrual pads to girls in schools [10,11,12,13,14,15].

As policies and programs to improve menstrual health increase in number and reach, surveys capturing population health or WASH access are called upon to monitor progress. Historic minimisation of female health concerns and silence around menstruation have resulted in scarce funding to support menstrual health research. Significant gaps in the evidence base mean there is little research available to inform best practices for monitoring menstrual health; there are currently no established indicators [16]. Surveys incorporating questions to monitor menstrual health have been informed by national policy objectives, developing international definitions (e.g., UNICEF [1]), and core components of menstrual health programming which have focused on improving access to commercial menstrual materials, as well as programs to improve sanitation infrastructure, and puberty education [1,5,17].

Three major population-based surveys conducted in LMICs—Performance Monitoring and Accountability 2020 (PMA), Multiple Indicator Cluster Survey (MICS), and the Demographic Health Survey (DHS)—have added menstruation-related questions in an effort to monitor menstrual health at the population level. PMA surveys include nine questions on women’s last menstrual period, including identifying the location and facilities used when changing menstrual materials, the materials used to collect or absorb menstrual blood, washing and drying practices for reusable materials; material disposal, and a question asking women if there is anything else they need to manage their periods that they do not usually have [18,19]. MICS contains four questions: women’s use of menstrual materials; whether they used reusable materials; whether they were able to wash and change in privacy while at home; and if there were any social activities, or school or work days that they did not attend during their last menstrual period [20,21]. The DHS contains four questions asking what women used to collect or absorb menstrual blood during their last menstrual period, whether they were able to wash and change in privacy while at home, and two knowledge-based questions on the menstrual cycle and pregnancy risk [22].

All three surveys include questions focused on the type of menstrual materials used. This focus is intuitive, reflects national policy emphasis, and is sensitive to the body of evidence reporting many women’s and girls’ dissatisfaction with their current materials. Further, it is aligned with quantitative studies on menstrual health, which most frequently report on material use as a key descriptor of menstrual health or hygiene. At the same time, monitoring the type of material used and reporting national percentages lends itself to interpretation of this data as an indicator of women and girls having their menstrual material needs met—a key aspect of menstrual health [23].

While population-based monitoring programs may objectively report menstrual material use and place no explicit judgement or value on which materials may be superior or represent menstrual health or hygiene status, the prevailing discourse around menstrual material use portrays commercially produced disposable menstrual pads as the “gold standard” menstrual material [10]. The use of menstrual pads to manage menstruation has often been considered to indicate adequate “menstrual hygiene”, with the distribution of these products the focus of many policy and non-governmental organization initiatives [10,24,25]. In this context, the use of menstrual pads may frequently be interpreted as an indication that populations’ menstrual material needs are being met, and specifically, this may be assumed to indicate that women have access to sufficient, preferred menstrual materials. There are likely to be limitations to this interpretation. First, different women may prefer different materials depending on their contexts, needs, and preferences [26]. Second, whether or not women use commercial menstrual pads does not capture the perceived quality of the products they use, if they have enough materials or the duration of wear, accessibility of menstrual products in their communities, or the cost of products relative to women’s household budgets, all of which may impact a person’s experience of menstruation. As a result, it is also possible that this indicator is differentially reflective of the experiences of different socio-demographic groups. For example, in a country such as Kenya, which has a high market penetration of menstrual pads, it is likely that women of both low and high socio-economic status (SES) report using pads to manage their menstruation; however, the nature of their pad use and their subsequent perceived menstrual needs may differ significantly. Thus, relying only on whether a person uses menstrual pads alone overlooks other factors, influenced by socio-demographics, that may determine whether a person’s material menstrual needs are met.

### The Present Study

The present study investigates the performance of menstrual pad use as an indicator of whether menstrual material needs are met at the population level. We undertake secondary analysis of publicly available PMA survey data from seven nationally representative samples, and five state or city representative samples from 10 countries collected between 2015–2017. We tabulate women’s self-reported material needs (needing clean materials, money, or access to a vendor) according to the type of menstrual material they report using in order to describe the extent to which menstrual pad use represents or misrepresents women’s reported menstrual material needs. Further, we assess whether any misrepresentation varies across socio-demographic groups. 

## 2. Materials and Methods 

We undertook secondary analyses of PMA data from 12 settings in 10 countries. PMA implements nationally or sub-nationally representative surveys with households and women in 13 settings across 11 countries in Africa and Asia [19]. Data are collected regularly—between every 6 weeks and every annum—through in-person interviews conducted via smartphone devices with real-time quality assurance and management, enabling rapid data collection and analysis.

The data used in this study are from surveys conducted in Uganda (between April–May 2017), Kenya (November–December 2016), Ethiopia (April–May 2017), Ghana (August–November 2016), Burkina Faso (November 2016–January 2017), Niger (February–April 2016), Lagos and Kaduna, Nigeria (August–September 2015), Kinshasa and Kongo Central, Democratic Republic of Congo (DRC, September–November 2017), Indonesia (October 2016–January 2017), and Rajasthan, India (October 2016–January 2017). With the exception of Kinshasa and Kongo Central, each site employed a multi-stage cluster sampling design in which enumeration areas (EAs) were drawn from a master sampling frame provided by a national or international statistical agency. After listing and mapping households in each EA, in each setting, a pre-set number of households per EA (ranging from 32–42) was randomly selected and invited to participate in the PMA household survey.

For each household that completed the survey, all females of reproductive age (15–49 years) were invited to participate in a female survey. Our sample includes women who completed the female survey who reported being usual members of the respective household, slept in the household the previous night, and reported menstruating in the previous three months. The final sample sizes after applying these criteria were 2709 in Uganda, 4364 in Kenya, 4784 in Ethiopia, 2837 in Ghana, 2158 in Burkina Faso, 1913 in Niger, 1169 in Lagos, 1993 in Kaduna, 2081 in Kinshasa, 1109 in Kongo Central, 8117 in Indonesia, and 5023 in Rajasthan.

We draw from menstrual health questions asked in the water, sanitation, and hygiene (WASH) module of female surveys, which inquired about the type of menstrual materials women used, the location(s) used to change materials, women’s perception of this location, menstrual material disposal practices, and self-reported unmet menstrual needs. Female surveys were conducted by trained female resident enumerators, and explicit and informed consent was obtained from participants before proceeding with interviews. All female interviews were conducted with auditory privacy and, when possible, visual privacy. More information on the PMA survey program is available from the PMA website [18]. 

### 2.1. Measures

#### 2.1.1. Menstrual Material Use

Menstrual material use was measured using the multiple-response question, “During your last menstrual period, what did you use to collect or absorb your menstrual blood?” Enumerators recorded all materials communicated by participants based on pre-set response options and probed for “anything else” to ensure they had captured all materials used. Response options for entry included: sanitary (menstrual) pads; tampons; cloth; cotton wool; toilet paper; paper from newspapers or books; natural materials, such as mud, dung, or leaves; foam from mattresses or other materials; a bucket (in Ethiopia only); no materials; or other. The pads response option included both disposable and reusable menstrual pads, and the cloth response option included newly purchased and repurposed cloths. Materials that enumerators categorized as “other” were not recorded.

Throughout this paper, we present two different approaches for operationalizing menstrual material use data. The first is through a grouped variable in which participants were coded to a single-response option. This variable separately coded those exclusively using a particular material and those using combinations of materials. Thus, the response options for this grouped single-response variable are mutually exclusive, enabling us to delineate, for example, between participants using menstrual pads exclusively from those using pads in combination with other items. We implemented the following response options: respondents using menstrual pads or tampons only (henceforth referred to as “pads only” due to the very low level of tampon usage); cloth only; cotton wool only; a combination of menstrual pads, cloth, or cotton wool; a combination of menstrual pads, cloth, cotton wool, and any other materials; or other (including no materials, natural materials, foam, paper, or “other” from the original question).

The second approach operationalized menstrual material use in its original form, as a multi-response variable. In this way, a single participant can be represented across several response categories and responses, and thus represent cases rather than participants. For example, a participant who reported using cloth and pads to manage their last period would be counted as a case under the binary variable cloth and, separately, the binary variable pads. With this approach, we gain insight into cases where a material was used “at all”, whether alone or in combination with other materials.

#### 2.1.2. Menstrual Material Needs

PMA included an open-ended multi-response question that asked, “Is there anything else that would help you manage your menstrual period that you do not usually have?” Again, participants provided a verbal response to this question which enumerators entered based on a pre-set selection of responses, and enumerators probed for additional needs by asking “anything else?” Pre-set options for enumerators to code against included: I have all I need, clean water, soap, clean absorbent materials, a private place, a safe place, knowledge, a place to buy clean absorbent materials, a place to dry used materials, a place to dispose of used materials, money, pain medication, or other. We selected response options directly related to having access to sufficient menstrual materials as indicative of unmet menstrual material needs—specifically, those who reported needing clean absorbent materials, a place to buy clean absorbent materials (a vendor), or money. Although women also have other needs, including spaces to change and dispose of materials, we restricted analyses to needs that were most likely to be misinterpreted as being met when women report using menstrual pads. We included money as a menstrual material need under the assumption that it may be used to purchase menstrual materials or reduce the financial burden of purchasing menstrual materials, though we acknowledge that participants may have cited needing money for other needs unrelated to menstrual materials, such as pain relief or soap. Based on these multi-response variables, we constructed a categorical menstrual needs variable with three options: clean menstrual material (which includes responses that involved any mention of clean absorbent as a need), money or vendor (which includes responses that involved any mention of money or vendor as needs), and no needs (which includes all other needs mentioned, as well as responses indicating the respondent had no additional menstrual needs). Respondents who were categorized as having “no needs” are referred to interchangeably as having no unmet menstrual material needs, or having their menstrual material needs met in this paper.

### 2.2. Analyses

Analyses were undertaken using Stata v15.1 (StataCorp, College Station, TX, USA). All analyses implemented setting-specific survey weights to accommodate the complex sampling design of each survey. Using descriptive statistics, we first calculated, for each setting, the proportion of women reporting use of each menstrual material based on both our categorical single-response variable, as well as our binary multi-response material use variables.

Next, we calculated, for a given menstrual material option, the proportion of users who reported needing materials, money, or a vendor to purchase materials, or those who had no material needs. We present this data using both the categorical single-response and binary multi-response menstrual material use variables, combining toilet paper and paper from newspapers or books into one “paper” multi-response category, and consolidating natural materials, foam, other, and no materials into one “other” multi-response category. We undertook inverse-variance weighted random-effects models to pool proportions across surveys using the *metapreg* module for Stata [27]. We used a random-effects model as we expected heterogeneity across surveys, and did not assume that the relationship between menstrual material use and unmet needs would be similar across countries. We present pooled estimates as percentages for ease of interpretation.

Lastly, we explored the usefulness of menstrual pad use as an indicator of women’s material menstrual needs being met across settings and across socio-demographic groups within each setting. We estimated the proportion of pad users who would be misrepresented as having their menstrual material needs met if we relied solely on their use of pads as an indicator. For this analysis, women were considered misrepresented if they reported using pads but also reported having any of the material needs defined above. For each setting, we provide the proportion of pad users who would be misrepresented across different socio-demographic characteristics, including the age group in 5-year intervals, highest level of education (none, primary, secondary, higher), wealth quintile, and rurality of household location (rural vs. urban). Wealth quintiles were calculated for each setting and thus are not comparable across settings. Rurality of household location was determined by relevant national or international statistical agencies for each setting. Again, proportions were pooled across surveys using a random-effects meta-analysis. We did this analysis both for women who reported using only pads based on our grouped single-response menstrual material use variable, as well as for women who reported using pads at all based on our binary multi-response material use variables to account for the different ways in which reporting agencies may use menstrual material data.

### 2.3. Ethical Approvals

This was a secondary analysis of existing publicly available data and was thus exempt from review by the Johns Hopkins University Bloomberg School of Public Health Institutional Review Board. Approvals for the human subjects research involved in the implementation of the PMA surveys were granted by local ethical review boards in each setting [18]. 

## 3. Results

### 3.1. Menstrual Material Use Across Settings

Table 1 displays the proportion of menstrual material use for each setting, with material use operationalized using both the single-response (pads only) and multi-response (pads) approaches. The top panel displays menstrual product use based on a grouped single-response variable, in which the percentages provided reflect the percent of respondents. For example, 52.2% of respondents in Uganda reported only using pads to manage their menses, and 30.6% reported only using cloth. The bottom panel displays menstrual product use as a multi-response variable in which product categories are not mutually exclusive. For example, 64.3% of respondents in Uganda reported using pads at all—whether alone or in combination with other items—to manage their last menstrual period, while 41.9% reported using cloth at all.

Based on the product use data provided in Table 1, the majority of respondents reported using pads in half of the locations—Uganda, Kenya, Ghana, Lagos, Kinshasa, and Indonesia—with exclusive pad users comprising the majority of respondents who reported using pads at all. Kenya, Ghana, Lagos, and Kinshasa had particularly high pad use, with a mean of 86.3% reporting using pads at all across each setting (range 84.0–89.3%), and a mean of 77.1% reporting using only pads (range 73.3–82.7%). Less than half of respondents reported using pads—alone or in combination with other items—in Ethiopia, Burkina Faso, Niger, Kaduna, Kongo Central, and Rajasthan, with over half of respondents reporting using only cloth in Burkina Faso, Niger, and Kaduna (range 55.7–59.8%).

Specific alternative materials were also popular in certain settings. Based on multi-response categories, use of cotton wool was relatively popular in Kenya (7.3%), Burkina Faso (15.4%), Niger (13.7%), and Kongo Central (17.1%), although only in Burkina Faso and Niger do those users appear to use cotton wool exclusively. Paper—tissue or newspaper—was also relatively common in Lagos, Kinshasa, and Kongo Central (15.3%–25.1%), while foam was also cited by a substantial percentage of people in Niger and Kongo Central (8.6% and 5.1%, respectively). Lastly, compared to other countries, a relatively high percentage reported using nothing to manage their menses in Ethiopia (11.1%), Niger (4.3%), and Burkina Faso (3.7%). Ethiopia was the only setting to include use of a bucket as an option for menstrual management, although only 0.2% reported using this method.

### 3.2. Menstrual Material Needs by the Type of Menstrual Material Used

Table 2 displays the pooled proportion and 95% confidence interval (CI) of respondents’ self-reported unmet menstrual material needs according to the type of menstrual material they used. This is displayed for material use defined using both single-response (pads only) and multi-response (pads) categorizations. This is presented by country in Appendix A.

Women who reported exclusively using pads during their last period generally had the highest proportion of met menstrual material need (e.g., “no needs”). Across surveys, the pooled proportion of exclusive pad users who reported having no additional material needs was 73.4% (survey range 35.7–96.5%, data shown in Figure 1 and Appendix A). While pads-only users reported the highest proportions of met menstrual material needs across the 12 settings, users of other materials also had relatively high proportions. Pooled across settings, 63.8% of cotton wool-only users had their menstrual needs met (range across surveys 41.6–100%, data shown in Appendix A). Cloth-only users and people who reported using a mix of products—pads, cloth, and cotton wool or pads, cloth, cotton wool, and other—tended to report the lowest proportions of met menstrual material needs, and the highest proportions of unmet menstrual material needs (e.g., “absorbent,” “money or vendor”). A pool of 49.2% (survey range 20.5–89.3%) of cloth-only users across countries reported having no additional needs, and a pool of 31.1% (survey range 7.5–62.0%) reported needing clean absorbent compared to a pool 14.1% (survey range 2.2–37.7%) of pad-only users. Cloth-only users and those who reported using a mix of products generally had the highest proportions of needing money or a vendor, though the overall reporting of money or vendors as a need was generally low relative to those who reported needing clean absorbent or having no needs across settings.

In addition to categorizing menstrual needs based on the grouped single-response menstrual product use categories, Table 2 also provides this information for the multi-response product use categories. In general, the multi-response categories exhibited lower met menstrual needs than their corresponding single-response categories, which is consistent with the finding that using product mixes tended to be associated with lower proportions of met menstrual material needs in the grouped single-response category.

A direct comparison of the single-response and multi-response approaches is provided in Figure 1, which displays the material needs of those who reported using pads only (based on the single-response approach) and those who reported using pads at all (based on the multi-response approach). Figure 2 similarly displays the material needs of those using cloth exclusively, and those who reported using any cloth.

### 3.3. Misrepresentaton of Material Needs by Socio-Demographic Group

We explored the performance of pad use as an indicator of menstrual material needs being met by calculating the proportion of users who would be misrepresented if pad use were to be interpreted as inferring that women had access to sufficient, preferred materials. Women were considered misrepresented if they reported using pads but still reported needing clean absorbent materials, money, or a vendor to manage their menses. Table 3 reports the proportion of exclusive pad users, defined using our single-response variable, who would be misrepresented for different demographic groups within each setting, as well as the pooled proportion. Table 4 presents the proportion of misrepresentation for demographic groups among those using pads, as defined by the multi-response variable approach.

Across settings, among those who reported exclusively using pads (“pads only”, Table 3), a pooled 26.4% of users (*I^2^* = 99.2, *p* <.001) would be misrepresented as having their material needs met based on their pad use. There was substantial variability across countries, with the misrepresented proportion ranging from 3.5% in Lagos to 63.3% in Kongo Central.

The proportion of menstrual pad users who were misrepresented as having their menstrual needs met based on pad use as an indicator varied meaningfully across demographic groups. Among those exclusively using pads, increasing disadvantage is associated with a greater proportion of that population being misrepresented. The largest differences were among those of different wealth quintiles. A pooled 16.8% of pad users in the highest wealth quintile across countries (survey range 2.6–58.3%) were misrepresented, compared to a pooled 45.9% of those in the lowest wealth quintile (survey range 8.5–100%). Similarly, a pooled 12.8% of pad users (survey range 2.9–50.9%) with higher than a secondary school education were misrepresented, compared to a pooled 31.4% who reported primary school as their highest level of education (survey range 3.0–65.0%). Among settings that included urban and rural variables, a pooled 38.5% of rural pad users were misrepresented (survey range 9.0–62.3%), compared to 17.1% of urban pad users (survey range 6.7–26.7%).

The overall proportion misrepresented for each setting increases by 0.3–8.6 percentage points when pad use is defined using a multi-response format that includes women who have used pads in combination with other products (see Figure 1, Table 2, and Appendix A). Demographic trends were similar when pad use was categorized using a multi-response approach (Table 3 and Table 4).

## 4. Discussion

This study assessed the performance of menstrual pad use as an indicator of menstrual health, using data from seven national-, and five city- or state-level PMA surveys conducted in 10 countries. Menstrual health programs aim to ensure that women and girls have access to sufficient quantities of clean and preferred materials [1], with increasing recognition that the type of material preferred may vary between contexts and individuals [28]. Our study found that countries differed in the proportion of the population using different types of menstrual materials, which provides useful information for understanding material use at the population level. However, the use of menstrual pads was not an accurate indicator of having access to sufficient materials and was particularly misrepresentative of the experiences of more disadvantaged women.

Menstrual material use differed across countries. Exclusive menstrual pad use ranged from 9.7% in Niger to 87.7% in Indonesia, and cloth use was more common in areas with low pad use. These data indicate large regional variations in menstrual material use practices. Differences may be attributable to cultural, historical, and economic variation, as well as market penetration of different products and national policies or standards applied to different menstrual products [10,29,30,31]. Representative national and regional menstrual material use data are likely to provide meaningful insights on the reach of national programs or market-based efforts to expand access to, or use of, certain products. Through providing insight on product use behavior, such data may also provide information that is beneficial in assessing water and waste management infrastructure needs related to the use of different menstrual materials. For example, reusable products require regular access to clean water for washing, while single-use products require resources related to disposal and waste stream management [26]. Governments, policymakers, NGOs with national-level programs, national utilities, and potentially also private sector product manufacturers and distributors are likely to benefit from this information.

Although the type of materials used provides meaningful population-level information, our study found that this was a poor indicator of women’s menstrual material needs being met, an indicator more closely aligned with objectives of menstrual health programmes. We found that across settings, respondents who reported exclusively using menstrual pads or cotton wool had higher proportions of met menstrual material needs. Those who reported using exclusively cloth or a mix of products more frequently reported needing more materials, money, or a vendor to manage their period. While these findings appear to support the narrative of menstrual pads as a “gold standard” menstrual material, other findings bring the use of pads as a population-based indicator of met menstrual need into question. Specifically, across settings, a substantial proportion of exclusive menstrual pad users reported needing additional clean menstrual materials, while almost half of cloth users reported having no additional menstrual material needs. These findings imply that menstrual pad use is not always indicative of menstrual material needs being met, nor is cloth use indicative of unmet menstrual material needs. Together, this information confirms the likely inadequacy of menstrual material use, and menstrual pad use in particular, as an indicator of menstrual health.

We estimate that use of menstrual pads as an indicator of met menstrual need would result in over one in four exclusive pad users being misrepresented as having their needs being met across the settings in this study. Use of this indicator would disproportionately impact disadvantaged women, with lower income, less educated, and rural women having the greatest proportion of misrepresentation. This means that if pad use were to be implemented as an indicator of met menstrual material need at the population level, we would disproportionately misrepresent these groups as having their needs met and subsequently risk overlooking their material needs. In the context of national or regional policy-making, or targeted NGO or multinational organization intervention planning, this may misinform the level of material need among particular groups and result in misallocated funding or under-estimates of the cost of implementing programs. This is particularly salient given that Hennegan et al. recently published a study demonstrating that PMA eligibility criteria for the menstruation module—which excludes women who have not menstruated in the past three months—differentially includes wealthy, urban, and educated women [32]. Taken together, these results suggest that implementing pad use alone as an indicator of met menstrual need has the potential to misrepresent the needs of disadvantaged groups whose practices and experiences are already underrepresented at the population level.

Misrepresentation also varied significantly across settings—populations that had high overall menstrual pad use tended to have lower misrepresentation, whereas populations with relatively high use of alternative materials—such as paper, foam, or unspecified materials—tended to have higher misrepresentation. Excluding regional surveys, countries that had lower misrepresentation also tended to be wealthier, with Indonesia and Ghana—the second and third lowest misrepresentations, behind Lagos—having Gross National Incomes of $3840 and $4650 per capita, respectively, versus $1970 and $1920 for Uganda and Burkina Faso, the countries with the third and fourth highest misrepresentations, behind Kinshasa and Kongo Central [33]. Given our findings of higher misrepresentation among disadvantaged groups, it is not surprising that lower income countries would have higher overall misrepresentation compared to relatively higher income countries. One potential explanation for the variation in misrepresentation across settings is that in higher income countries, women may be more likely to have adequate quantities of menstrual pads and be able to purchase higher quality pads with less burden on their household resources, while in lower income countries, women may have fewer pads to use or experience greater strain accessing and purchasing these materials.

Throughout this paper, we presented two approaches for categorizing multi-response menstrual material use data. The first was through a grouped single-response variable that enabled us to identify users who exclusively used a given material, versus those who used combinations of materials. The second approach was through the original binary multi-response variables that indicated if people used a material at all, whether alone or in combination with other products. These approaches were selected to represent the most likely ways that population-level data would be disseminated, such as in online briefings, reports, or aggregated data provided to policy makers or to the public. Trends were generally aligned between the two approaches. While the multi-response menstrual product use variables were beneficial in indicating overall use of a given material within a population, categorizing menstrual pad use in this way consistently resulted in higher overall misrepresentation of menstrual needs as being met. Surveys with multi-response data should consider, and perhaps report on, multiple approaches for categorizing menstrual material use data to convey the different information each approach provides, as well as to minimize the possibility of misrepresentation and misinterpretation of menstrual needs among a population. Further, survey administrators should consider the information that may be gained or lost by implementing a multi- versus single-response question to participants. In our case, PMA’s use of a multi-response question enabled us to code menstrual material use in both multi- and single-response formats. 

This study leveraged existing PMA survey data to investigate our research question. While this allowed us to compare results across 10 different countries, the secondary analysis of data meant we were limited by existing survey questions and data. PMA surveys had a single response option for pads, including both disposables and reusables. Although the vast majority of pad users are likely to be using disposable pads, we were unable to separate those using reusable pads. Respondents were asked to identify their own unmet menstrual needs through an open-ended question that was coded into a menstrual needs variable. It is plausible that there was variation in participant responses between individuals and across contexts. The open-ended nature of the menstrual needs question, and its position near the end of the survey may have resulted in overall underreporting of menstrual needs compared to close-ended questions. Unmet needs were recalled following a short series of questions on menstrual hygiene, and women may have been primed to consider the needs they had already been asked about, rather than those not included in the survey. It is also possible that respondents underreported needs that were not discussed in the survey, such as access to menstrual material vendors. Participants may also have provided socially desirable responses and the detail in their response may have been shaped by comfort with the interviewer and personal historical experiences of menstrual management. Perceived menstrual needs are also likely to be influenced by a variety of factors, such as community expectations for menstrual management and exposure to commercial menstrual materials or advertising [3,24,25]. Finally, we recognize that women have a range of needs related to their menstrual management, such as washing and drying reusable materials or disposing of single use products. These needs were beyond the scope of this analysis, as we focused on unmet needs that are likely to be overlooked by the use of menstrual material type as an indicator for menstrual health. Other questions will be needed in national surveys so as to be sensitive to broader needs.

Despite these limitations, this study has meaningful implications for those who plan and monitor menstrual health programs at the regional or national level, particularly in LMICs. We demonstrate that menstrual material use varies across settings and that no menstrual material is consistently associated with material needs being fully met or unmet. These findings are consistent with ongoing efforts within menstrual health research and practice to focus on community needs and individual choice in menstrual material selection, rather than providing a “one-size fits all” solution [1,10,11,34], as well as recent efforts to produce more female-centered measures for menstrual experience and needs [28]. Drawing on this work, indicators capturing whether women have access to enough, and their preferred menstrual materials may provide national data that is more closely aligned with the goals of menstrual health programmes and policies. While menstrual material use may be used to measure penetration or uptake of national-level product access initiatives, we show that menstrual material use is not indicative of met material needs, particularly for disadvantaged groups. Efforts must be made to identify or develop accurate indicators of menstrual health at the population level that are reflective of menstruators’ experiences and an agreed-upon definition of menstrual health. 

## 5. Conclusions

The inclusion of indicators for menstrual health is an essential step forward in adequately monitoring women’s health and WASH needs, and assessing the progress made towards improved menstrual experiences. Data collected to date also present an opportunity to critically appraise the performance of the questions currently used, to determine whether these are best placed to capture menstrual health, or if modifications are needed. We demonstrate that menstrual material use is not an indicator of menstrual material need, and further, that relying intentionally or by default on menstrual pad use as an indicator of met menstrual material needs disproportionately misrepresents the experiences of low-income, less educated, and rural pad users. Investment is warranted to identify the most appropriate and highest-priority indicators of menstrual health at the population level. Without a concerted effort to implement accurate indicators, we risk misrepresenting, and thus overlooking, the needs and experiences of the most disadvantaged groups.

## Figures and Tables

**Figure 1 ijerph-17-02633-f001:**
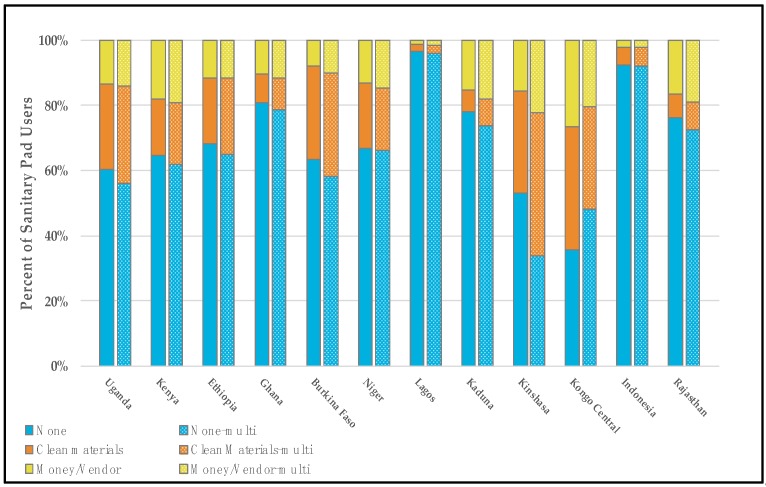
Reported menstrual material needs of menstrual pad users whose pad use was categorized using a single-response variable denoting exclusive pad use (left) and a multi-response variable denoting any pad use (right, patterned). Response options included no material needs (blue, bottom), clean materials (orange, middle), and money/vendor (yellow, top).

**Figure 2 ijerph-17-02633-f002:**
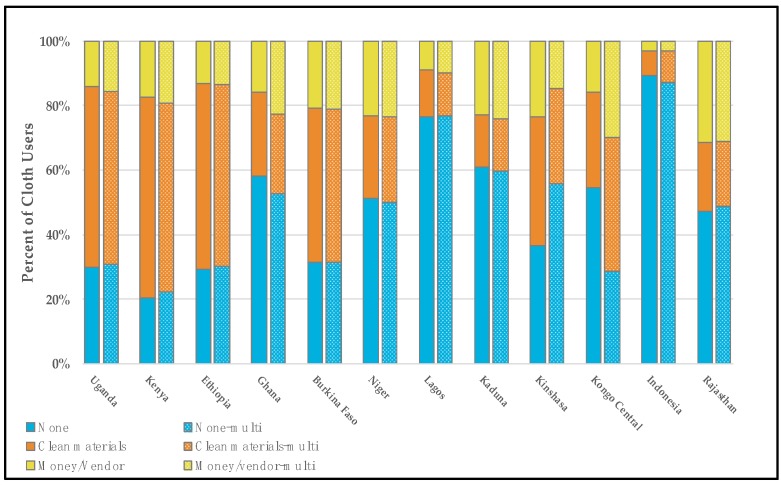
Reported menstrual material needs of cloth users whose cloth use was categorized using a single-response variable denoting exclusive cloth use (left) and a multi-response variable denoting any cloth use (right, patterned). Response options included no material needs (blue, bottom), clean materials (orange, middle), and money/vendor (yellow, top).

**Table 1 ijerph-17-02633-t001:** Reported menstrual material use by country. Material use is categorized using both a single-response and a multi-response variable.

Menstrual Material	Uganda	Kenya	Ethiopia	Ghana	Burkina Faso	Niger	Lagos, Nigeria	Kaduna, Nigeria	Kinshasa, DRC	Kongo Central, DRC	Indonesia	Rajasthan
	(*n* = 2709)	(*n* = 4364)	(*n* = 4784)	(*n* = 2837)	(*n* = 2158)	(*n* = 1913)	(*n* = 1169)	(*n* = 1993)	(*n* = 2081)	(*n* = 1109)	(*n* = 8117)	(*n* = 5023)
**Grouped single-response material categories (% of respondents)**												
Pads only	52.2	77.5	34.0	82.7	13.6	9.7	74.7	27.4	73.3	20.8	87.7	39.1
Cloth only	30.6	9.0	38.4	8.6	58.0	59.8	4.0	55.7	3.6	29.5	7.3	49.0
Cotton wool only	1.6	2.5	1.2	0.3	13.2	12.1	1.2	0.3	0.8	6.8	0.1	0.3
Pads, cloth, cotton wool	12.5	7.8	7.0	3.4	4.1	1.8	2.6	8.9	1.9	8.3	3.0	10.4
Pads, cloth, cotton wool, other	1.2	2.3	2.5	3.4	1.8	2.5	9.5	4.4	11.1	15.4	0.2	0.2
Other ^1^	2.0	1.0	16.9	1.5	9.3	14.2	8.2	3.4	9.3	19.2	1.8	1.0
**Multi-response material categories (% of cases)**												
Pads	64.3	86.1	39.8	89.3	15.8	9.9	85.5	37.4	84.0	33.9	90.7	49.2
Tampons	0.2	1.2	0.3	1.2	0.8	0.9	0.6	0.0	1.9	5.6	0.1	0.1
Cloth	41.9	14.1	47.3	12.0	63.1	63.6	6.5	67.6	7.4	40.4	10.3	59.4
Cotton wool	5.1	7.3	3.0	1.0	15.4	13.7	2.6	1.2	2.2	17.1	0.3	1.4
Paper	0.6	2.2	0.2	3.3	3.9	0.2	17.1	4.6	15.3	25.1	0.2	0.1
Foam	0.4	0.5	0.8	0.3	1.2	8.6	0.5	0.34	0.5	5.1	0.0	0.1
Natural materials	0.3	0.1	2.2	0.6	0.1	0.0	0.0	0.0	0.0	0.0	0.0	0.1
None	1.3	0.4	11.1	0.1	3.7	4.3	0.0	1.1	0.3	2.7	1.7	0.9
Bucket			0.2									
Other ^2^	1.2	0.0	4.9	0.8	2.3	3.6	0.1	1.7	4.5	2.7	0.1	0.1

^1^ Includes foam, natural materials, bucket, paper, and “other” from the original question, or any combination of those materials, and no materials. ^2^ Materials marked as “other” from the original question.

**Table 2 ijerph-17-02633-t002:** Pooled proportion of menstrual material needs by the type of menstrual material used.

Menstrual Material	Self-Reported Menstrual Material Needs Pooled Proportion (as Percent) (95% CI)		
	None	Absorbent	Money or Vendor
**Menstrual Material (grouped)**			
Pads only	73.4 (61.2–82.8)	14.1 (8.7–22.0)	10.4 (6.6–16.0)
Cloth only	49.2 (36.3–62.1)	31.1 (21.4–42.8)	16.3 (11.9–21.8)
Cotton wool only	63.8 (52.8–73.5)	24.2 (17.5–32.4)	14.2 (10.8–18.4)
Pads, cloth, cotton wool	50.6 (40.8–60.3)	25.9 (17.9–35.9)	19.1 (13.4–26.6)
Pads, cloth, cotton wool, other	47.6 (30.4–65.4)	30.8 (20.2–44.0)	16.9 (10.2–26.9)
Other ^1^	61.4 (47.5–73.6)	27.0 (17.4–39.3)	9.2 (6.2–13.5)
**Menstrual Material (multi-response)**			
Pads	70.6 (57.5–80.9)	15.6 (9.6–24.3)	11.5 (7.4–17.5)
Cloth	48.1 (35.9–60.6)	30.7 (21.6–41.7)	17.3 (12.6–23.4)
Cotton wool	64.8 (51.8–75.9)	20.6 (13.7–29.9)	14.5 (10.1–20.5)
Paper	55.2 (29.6–78.2)	25.1 (14.6–39.6)	14.9 (7.7–26.9)
All others ^2^	53.0 (41.3–64.5)	33.6 (24.1–44.6)	11.7 (8.6–15.7)

^1^ Includes foam, natural materials, bucket, paper, and “other” from the original question, or any combination of those materials, and no materials. ^2^ Includes foam, natural materials, no materials, bucket, and responses originally recorded as other.

**Table 3 ijerph-17-02633-t003:** Misrepresentation of menstrual needs as being met among exclusive menstrual pad users (single-response variable), by socio-demographic characteristics.

Socio-Demo- Graphic Group	Uganda	Kenya	Ethiopia	Ghana	Burkina Faso	Niger	Lagos, Nigeria	Kaduna, Nigeria	Kinshasa, DRC	Kongo Central, DRC	Indonesia	Rajasthan	Pooled Proportion
	(*n* = 1415)	(*n* = 3359)	(*n* = 2533)	(*n* = 23,356)	(*n* = 446)	(*n* = 403)	(*n* = 871)	(*n* = 595)	(*n* = 1624)	(*n* = 367)	(*n* = 7143)	(*n* = 1840)	
	%	%	%	%	%	%	%	%	%	%	%	%	(95% CI)
**Age**													
15–19	40.8	38.8	37.2	21.0	46.7	34.9	4.3	29.4	50.9	62.4	9.6	31.6	26.5 (15.2–42.1)
20–24	34.3	33.4	29.0	19.6	31.8	45.6	2.1	22.2	46.6	72.1	7.8	27.5	27.2 (17.2–40.3)
25–34	38.9	34.4	28.5	19.7	41.8	26.3	2.8	16.6	43.6	70.4	8.1	19.2	25.0 (15.4–37.9)
35+	46.0	34.3	26.7	17.4	18.0	16.2	4.4	17.0	47.0	49.8	6.5	11.7	20.8 (13.3–31.2)
**Education**													
None	49.5	40.6	43.4	31.1	57.8	47.9	0.0	25.4	54.7	49.7	13.0	29.7	35.6 (26.7–45.5)
Primary	51.9	42.3	40.6	29.0	38.7	37.6	3.0	13.6	49.2	65.0	11.1	32.6	31.4 (20.9–44.2)
Secondary	32.2	34.7	25.5	18.2	31.9	31.5	4.1	24.8	48.0	65.3	7.4	24.6	25.6 (16.7–37.1)
Higher	17.2	17.9	9.6	3.1	19.4	9.9	2.9	18.3	40.4	50.9	3.5	13.2	12.8 (7.4–21.3)
**Wealth**													
1 (lowest)	71.3	53.0	62.5	41.3	21.0	100	8.5	31.0	61.0	65.5	14.6	50.6	45.9 (29.2–63.6)
2	56.1	45.3	41.2	20.8	66.1	96.7	6.0	29.4	51.2	89.7	11.3	40.6	43.6 (25.6–63.5)
3	42.3	41.0	52.8	18.0	58.5	77.1	2.1	34.3	49.5	60.8	9.4	34.6	35.3 (21.0–5.29)
4	33.6	33.4	43.6	16.3	62.4	42.4	1.1	21.0	38.9	67.8	4.9	22.2	26.2 (14.3–42.9)
5 (highest)	24.1	16.9	21.9	6.1	20.5	25.6	2.6	19.6	44.7	58.3	3.1	14.2	16.8 (9.7–27.5)
**Rurality**													
Urban	25.7	25.9	22.6	13.4	26.7	22.7		9.3			6.7	13.4	17.1 (12.4–23.0)
Rural	46.1	40.9	46.9	28.3	62.3	58.9		37.7			9	35.8	38.5 (27.3–51.1)
**TOTAL**	**39.2**	**35.4**	**31.4**	**19.4**	**36.5**	**33.2**	**3.5**	**21.9**	**47.0**	**63.3**	**7.7**	**18.8**	**26.4 (17.1–38.5)**

**Table 4 ijerph-17-02633-t004:** Misrepresentation of menstrual needs as being met among those using any menstrual pads (multi-response variable), by socio-demographic characteristics.

Socio- Demo-Graphic Group	Uganda	Kenya	Ethiopia	Ghana	Burkina Faso	Niger	Lagos, Nigeria	Kaduna, Nigeria	Kinshasa, DRC	Kongo Central, DRC	Indonesia	Rajasthan	Pooled Proportion
	(*n* = 1737)	(*n* = 3767)	(*n* = 2816)	(*n* = 2542)	(n = 526)	(*n* = 451)	(*n* = 997)	(*n* = 788)	(*n* = 1778)	(*n* = 535)	(*n* = 7403)	(*n* = 2341)	
	%	%	%	%	%	%	%	%	%	%	%	%	(95% CI)
**Age**													
15–19	45.5	41.0	42.4	23.2	53.3	36.4	4.8	34.9	56.2	66.7	9.8	34.8	34.2 (23.1–47.2)
20–24	38.0	35.1	32.2	20.6	34.1	46.1	2.3	22.9	52.0	69.4	8.2	30.9	28.6 (17.9–42.4)
25–34	43.6	37.8	30.7	22.0	45.2	27.7	3.4	22.1	49.4	69.8	8.2	22.3	28.0 (17.7–41.2)
35+	49.3	37.9	28.8	19.8	26.3	16.6	4.7	20.9	49.3	57.9	7.1	18.6	23.2 (15.3–33.5)
**Education**													
None	49.4	44.9	38.9	34.6	56.6	48.2	6.1	28.1	50.1	49.4	13.0	31.5	37.3 (29.5–45.9)
Primary	56.7	45.7	44.7	31.2	47.5	36.1	2.4	26.3	59.0	66.3	11.8	36.3	35.6 (23.7–49.6)
Secondary	35.4	37.0	28.8	19.8	38.2	32.3	4.3	27.4	51.8	69.1	7.5	28.0	28.0 (18.1–40.6)
Higher	19.4	18.7	11.2	4.0	19.8	13.6	3.8	22.6	43.8	46.9	3.8	15.8	14.4 (8.7–23.0)
**Wealth**													
1 (lowest)	73.1	55.4	68.2	44.8	45.6	82.7	8.5	43.0	65.3	70.2	15.0	47.5	49.6 (34.8–64.5)
2	60.9	48.9	44.2	22.2	68.7	96.7	6.2	43.2	54.3	79.2	11.8	40.6	44.8 (28.1–62.7)
3	47.8	43.9	56.6	20.3	66.3	78.4	2.7	46.9	55.8	55.4	9.7	38.2	38.5 (23.7–55.9)
4	38.8	36.0	44.5	17.8	59.2	44.6	2.2	25.8	42.1	67.3	5.1	27.5	29.0 (17.1–44.6)
5 (highest)	24.7	17.6	23.9	6.3	32.32	26.4	2.6	19.1	50.4	63.6	3.1	15.4	18.5 (10.3–30.8)
**Rurality**													
Urban	27.8	26.8	24.7	14.7	29.0	25.0		9.7			6.7	15.6	18.4 (13.3–25.0)
Rural	50.8	44.4	49.5	31.3	67.5	58.9		41.9			9.6	39.0	41.7 (29.9–54.6)
**TOTAL**	**43.7**	**38.2**	**35.0**	**21.4**	**41.6**	**34.0**	**3.9**	**26.3**	**51.8**	**65.7**	**8.0**	**27.38**	**29.3 (19.0–42.2)**

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
