# Peer review of "National Monitoring for Menstrual Health and Hygiene: Is the Type of Menstrual Material Used Indicative of Needs Across 10 Countries?"

_ijerph, 2020, doi:10.3390/ijerph17082633_

Round 1
Reviewer 1 Report
This manuscript is very informative and timely to force people to think about menstruation and unmet needs in an environment that is already prejudiced towards disposable menstrual pads, without questions on availability or quality or disposal. It is excellent written and is a great fit for this journal on the intersection of public health and environmental research. The authors present two different ways of analysing the data and show that results would be similar; a strength of this paper. The discussion covers my greatest concern, that the unmet needs were based on open questions in an established survey, whereby respondents may not have provided the needed information compared to closed questions. Perhaps the authors can add to the discussion what direction the limitations may have biased their results (e.g. underestimate?). And perhaps the authors can suggest indicators that they think are suitable to monitor progress in MHM. E.g. would adding the frequency of need of change of the item in addition to type of item be a better indicator? How can one capture the quality of a pad?
I have otherwise few suggestions or questions.
Methods: line 145. Response options: these did not include menstrual cups and cloths. Were menstrual cups mentioned at all in the “other” responses? Cloth was not a separate category? Was it possible to discriminate between home-made cloths and reusable commercially available sanitary pads?
Figure 1 and 2: what is your conclusion from comparing these differently defined outcomes?
Table 1 probably represents untold suffering by millions of women in these countries, making do with whatever is available, and hidden pollution of all the pads that need to be disposed of. It would be interesting to know more about the “bucket method” in Ethiopia, but that may be beyond the scope of this article. Ethiopia and Kinshasa (and Niger) still have a considerable proportion other in the multi-response material category; perhaps as a footnote it can be added what this “other” options are, if known.
Discussion: “Excluding regional surveys, countries that had lower misrepresentation also tended to be wealthier, with Indonesia and Ghana—the second and third lowest misrepresentation behind Lagos—having Gross National Incomes of $3,840 and $4,650 per capita, respectively, versus $1,970 and $1,920 for Uganda and Burkina Faso, the countries with third and fourth highest misrepresentation behind Kinshasa and Kongo Central.” Please add the reference from where this information was obtained.
In conclusion: Please add “.” to the following sentence: “We demonstrate that menstrual material use is not an indicator of menstrual material need, and further, that relying intentionally or by default on sanitary pad use as an indicator of met menstrual material needs disproportionately misrepresents the experiences of low-income, less educated, and rural pad users Investment is warranted to identify the most appropriate and highest priority indicators of menstrual health at the population level.”
Reviewer 2 Report
The study is well considered and presented, and is poised to make strong contributions to the field of menstrual health and hygiene.
I do not require major revisions before recommending it for publication, but I do wonder if the authors might consider addressing more clearly their decision to only include the 3 response options they did as indicative of "unmet menstrual material needs." If a place to buy material (vendor) and money are both included, what is the authors' justification for leaving out the need for a place to dry used materials, a place to dispose, etc. While these might be conceptualized as distinct from the sub-domain of "obtaining menstrual materials," they may still be just as crucially connected to a woman's ability to use her preferred method at all. A woman may want to use reusables but knowing she would not have a way to wash and dry them may prevent her from using her preferred material. Such an individual could then be classified as using a disposable menstrual pad, even though she is not using the material she would most prefer to use because her other needs were not met in order to make her preferred choice possible. I am not necessarily suggesting the authors change their analyses, but perhaps more clarity on this issue would be helpful for readers.
The following are suggested minor revisions:
- Line 18: apostrophe is missing on "respondents"
- Line 42 and 43 refer to "menstrual products," whereas line 55 refers to "commercial menstrual materials." Is this difference meaningful? If not, there might be benefit in keeping the terminology consistent.
- Line 64 is missing a period/full-stop
- Line 82: Reconsider word choice (use of "evoked") to improve sentence clarity
- Line 84 switches to using the term "commercial sanitary pads" when up to this point in the paper, the authors have used the term "disposable menstrual pads." Consider remaining consistent in the terminology used, unless the switch was intentional. If the switch was intentional, the reason has not been made clear to the reader
- Line 131: is Water, Sanitation, and "Health" correct?
- Line 145: It appears that the response option of "cloth" might be missing in this sentence
- Third para in the Discussion section: end of second line in the para-- the word "and" should be "an"
- Last line of page 13: remove the contraction (haven't)
- First line of page 14: "menstrual product selection"...why not remain consistent with the use of the word "material"?
- 4th line up from the bottom in the Conclusion section is missing a period/full-stop
